# Synthesis of Berberine and Canagliflozin Chimera and Investigation into New Antibacterial Activity and Mechanisms

**DOI:** 10.3390/molecules27092948

**Published:** 2022-05-05

**Authors:** Wenhui Hao, Shiying Che, Jinsheng Li, Jingyi Luo, Wanqiu Zhang, Yang Chen, Zijian Zhao, Hao Wei, Weidong Xie

**Affiliations:** 1State Key Laboratory of Chemical Oncogenomics, Shenzhen International Graduate School, Tsinghua University, Shenzhen 518055, China; haowh20@mails.tsinghua.edu.cn (W.H.); luojy19@mails.tsinghua.edu.cn (J.L.); wanqiuzhangx@126.com (W.Z.); 15555373590@163.com (Y.C.); 2School of Chemistry and Materials Science, Huaihua University, Huaihua 418000, China; chesysclyx@163.com (S.C.); jsli1006@163.com (J.L.); 3College of Pharmacy, Shaanxi University of Chinese Medicine, Xi’an-Xianyang New Ecomic Zone, Xianyang 712046, China; 4Key Laboratory of Research and Utilization of Ethnomedicinal Plant Resources of Hunan Province, Huaihua University, Huaihua 418000, China; 5Shenzhen Key Laboratory of Health Science and Technology, Institute of Biopharmaceutical and Health Engineering, Shenzhen International Graduate School, Tsinghua University, Shenzhen 518055, China

**Keywords:** berberine, canagliflozin, antibacterial activity

## Abstract

Berberine is an isoquinoline alkaloid isolated from Chinese herbal medicines such as Coptis chinensis. It has many pharmacological actions, such as antibacterial, hypoglycemic, anti-inflammatory, and so on. However, due to the low lipophilicity of berberine, it is difficult to penetrate the bacterial cell membrane and also difficult to be absorbed orally and usually needs a relatively high dose to achieve the ideal effect. The purpose of this study is to transform the structure of berberine in order to improve the bioavailability of berberine and reduce the dosage. Moreover, we introduce a pharmacophore named Canagliflozin, a hypoglycemic drug (which was also found to have potential anti-bacterial activity) into BBR to see whether this new compound has more existed activities. We at first connected berberine with Canagliflozin, to form a new compound (BC) and see whether BC has synergic effects. We use microbroth dilution method to determine the minimum inhibitory concentration of BC, determine the bacterial growth with the enzyme labeling instrument, observe the formation of bacterial biofilm with crystal violet staining method, observe the bacterial morphology with field emission scanning electron microscope, and determine the intracellular protein with SDS-PAGE. The above indicators reflect the damage of BC to bacteria. New compound BC was successfully obtained by chemical synthesis. The minimal inhibitory concentration of compound BC on three bacteria was significantly better than that of berberine and canagliflozin alone and the combination of berberine and canagliflozin. Moreover, compound BC has obvious destructive effect on bacterial morphology and biofilm, and the compound also has destructive effect on intracellular proteins. Therefore, new compound BC has broad-spectrum antibacterial activity and the inhibitory effect of BC might play a role by destroying the integrity of biofilm and the intracellular protein of bacteria. In conclusion, we create a new molecular entity of berberine and Canagliflozin chimera and open up a new prospect for berberine derivatives in the treatment of bacterial infection.

## 1. Introduction

In recent years, bacterial drug resistance has become a major problem in the clinical treatment of bacteria in various countries around the world. Many superbacteria and super drug-resistant bacteria spread from one country to another at a very fast speed, which is closely related to the wide use of antibiotics in the past 80 years [1]. At present, about 700,000 people worldwide die from infection with drug-resistant bacteria every year, and it is predicted that it may increase to 10 million per year by 2050 [2]. Therefore, it is urgent to find a new treatment scheme for bacterial infection that can replace antibiotics.

Berberine (BBR, C_20_H_19_NO_5_, MW 336.37, Figure 1) is an isoquinoline alkaloid and positively charged particle isolated from Chinese herbal medicines such as Coptis chinensis (Figure 1). It has many pharmacological activities such as antibacterial [3], hypoglycemic [4], and anti-inflammatory [5]. However, due to the poor lipophilicity of BBR, it is difficult to penetrate the bacterial cell membrane and difficult to be absorbed orally [6]. BBR needs a relatively high dose to achieve the ideal effect, which limits its clinical application. At present, there have been some reports on the transformation of general molecules into BBR derivatives can have better activities BBR [7]. Therefore, changing the structural characteristics of BBR and then changing its antibacterial activity through molecular structure transformation is one of the important directions of BBR transformation.

Canagliflozin (CAN, Figure 1) is a sodium glucose cotransporter 2 inhibitor (SGLT2), which can reduce the absorption of glucose by the kidney, so as to achieve the hypoglycemic effect. More and more studies show that CAN also has many other activities. In our previous studies, CAN exerts anti-inflammatory [8,9] and anti-tumor [10,11] effects by regulating glucose metabolism and autophagy. In addition, CAN regulates intestinal microorganisms and change the composition of intestinal microorganisms [12]. Therefore, CAN may have a potential effect on bacterial microorganisms. CAN may have other interesting pharmacophores besides SGLT2 and are worth investigating further.

In view of the similarity between BBR and CAN on bacterial regulating activities, this study modified the structure of BBR and CAN, connected the two compounds to form a new compound, and tested the efficacy of the new compound. Considering that both BBR and CAN have separate hypoglycemic activities. In the preliminary study, we have thought this new compound sourcing from two compounds might have additional or enhanced hypoglycemic activities. Diabetes is usually associated with an increase in bacterial infection [13]. This new compound was also expected to have anti-diabetic and anti-bacterial activities since we did not have ideal drugs to treat diabetic infections. Unexpectedly, after checking those activities, we did not find that this new compound has significant hypoglycemic activity as CAN did (data not shown). Despite this, we found that this new compound has stronger antibacterial effects than BBR alone or BBR in combination with CAN, and further discussed the antibacterial mechanism of this compound.

## 2. Materials and Methods

### 2.1. Materials

BBR and CAN were obtained from Leyan reagent Co., Ltd., Shanghai Haohong Biomedical Technology Co., Ltd., Shanghai, China. Nutrient broth medium powder (HB0108-4) and modified medium solution (HBYM329) were obtained from Hope Bio-Technology Co., Ltd., Qingdao, China. Agar (AB0016H) was purchased from Sangon Biotech Co., Ltd., Shanghai, China. *Staphylococcus aureus, Pseudomonas aeruginosa and Escherichia coli* were obtained from Beina Chuanglian Biotechnology Research Institute, Beijing, China. *Pseudomonas aeruginos* and *Escherichia coli* are Gram-negative bacteria, and *Staphylococcus aureus* is Gram-positive bacteria. Bacterial protein extraction kit (BC3750-50T) was purchased Solarbio, Beijing, China. PAGE Gel Fast Preparation Kit (PG113) was purchased from Epizyme Biomedical Technology Co., Shanghai, China and Brilliant Blue R250 (ST1123-25g) was purchased from Beyotime, Shanghai, China. Electron microscope fixative (G1102-100mL) was purchased Service bio, Wuhan, China.

### 2.2. Compound Synthesis

All product mixtures were analyzed by thin layer chromatography glass-backed silica (TLC) plates with a fluorescent indicator from Branch of Qingdao Haiyang Chemical Co., Ltd., Qingdao, China. UV-active compounds were detected with a UV lamp (λ = 254 nm, 365 nm). For flash column chromatography, silica gel (200–300 mesh) was used as stationary phase. ^1^H and ^13^C NMR spectra were recorded on a Varian INOVA-400 in deuterated chloroform and deuterated methanol at 25 °C with residue solvent peaks as internal standards (δ = 7.26 ppm for ^1^H-NMR and δ = 77.16 ppm for ^13^C-NMR in deuterated chloroform and δ = 3.31 ppm for ^1^H-NMR and δ = 49.00 ppm for ^13^C-NMR in deuterated methanol). Chemical shifts (δ) are reported in ppm, and spin-spin coupling constants (J) are given in Hz, whereas multiplicities are abbreviated by s (singlet), d (doublet), t (triplet), q (quartet) and m (multiplet). Mass spectra were recorded on a ThermoFinnigan MAT95XP microspectrometer and High-resolution mass spectra (HRMS) were recorded on Agilent Technologies Accurate Mass Q-TOF 6530 microspectrometer. Melting points were recorded on a national standard melting point apparatus (Model: Taike XT-4) and were uncorrected. The mass spectroscopy measurements were operated in positive ion mode with the scan, working under the following conditions: the full-scan mass spectra were recorded from *m*/*z* 100 to *m*/*z* 1500; capillary voltage: 3.0 V; cone voltage: 40 kV; extractor voltage: 3.0 kV; source temperature: 120 °C, desolvation temperature: 400 °C; desolvation gas (nitrogen): 700 L/h; cone gas (nitrogen): 50 L/h.

### 2.3. Synthesis of Dihydroberberine

Under the air atmosphere, a Schlenk tube (35 mL) equipped with a magnetic bar was prepared for the following reaction. Berberine hydrochloride (1 mmol, 371.8 mg) was dissolved in 5 mL pyridine. NaBH_4_ (2 mmol, 75.6 mg) was added to the mixture. Then, the reaction mixture was allowed to stir at room temperature for 1 h (Figure 2). The mixture was washed with water (15 mL × 3) and then dried under vacuum overnight to give the corresponding dihydroberberine [14,15].

### 2.4. Synthesis of CAN Bromide

Under the air atmosphere, a Schlenk tube (35 mL) equipped with a magnetic bar was prepared for the following reaction. CAN (1 mmol, 444.5 mg) was dissolved in 5 mL anhydrous DMF solution, then added into NBS (2.5 mmol, 445 mg) and PPh_3_ (3.5 mmol, 917.7 mg) under cooling condition, the mixture was allowed to stir at 50 °C for 4 h (Figure 3). After cooling to room temperature, methanol was added to destroy excess reagent. N-butyl alcohol was added to remove DMF. The filtrate was concentrated, ether was extracted, and the oily crude product was purified by column chromatography using silica gel (200–300 mesh) as stationary phase and a mixture of methylene chloride and methanol as eluent (40:1) to give the corresponding CAN bromide [16,17,18,19,20,21,22].

### 2.5. Synthesis of BC

Under the argon atmosphere, a Schlenk tube (35 mL) equipped with a magnetic bar was prepared for the following reaction. Dihydroberberine (1.5 mmol, 505 mg) and CAN bromide (1.0 mmol, 506 mg) was dissolved in 2 mL acetonitrile. Then, NaI (0.2 mmol, 30 mg) was added dropwise to the system and the mixture was allowed to stir at 130 °C for 18 h (Figure 4). After cooling to room temperature, the mixture was purified by column chromatography using silica gel (200–300 mesh) as stationary phase and a mixture of methylene chloride, methanol and triethylamine as an eluent (1:1:0.02) to give the corresponding BC [23,24].

### 2.6. Determination of Minimum Inhibitory Concentrations 80 (MIC80)

According to the relevant standards of American Society for clinical laboratory standardization [25,26], the MIC80 values of different drugs were detected by microbroth dilution method. Bacterial strain activated in nutrient broth at 37 °C were centrifuged (8000× *g*, 5 min, 4 °C). After centrifugation wash twice with PBS, the bacterial pellets were resuspended in nutrient broth. Then, the absorbance of bacterial suspension was adjusted to OD600 = 0.5. The bacterial suspension was then diluted 400× in nutrient broth to a final cell concentration of 5 × 10^5^ colony-forming units CFU/mL. CAN, BBR, BBR+CAN (B+C), BC dissolved in DMSO were added to bacterial suspension to achieve final concentrations of 0.1, 0.2, 0.3, 0.4, 0.5, 0.6 and 0.7 mM. The final concentration of DMSO was 0.1% (*v*/*v*). Then, 200 μL of the culture was transferred into each well on 96-well microtiter plates. Samples were incubated at 37 °C for 24 h and the OD600 values were determined after 24 h using a microplate reader (Epoch; Biotake, Beijing, China). The MIC80 was the concentration of drugs with antibacterial rate of more than 80%.

### 2.7. Growth Curve

Strains were grown to OD620 = 0.5 in nutrient broth medium and then diluted in 100-fold nutrient broth medium to a final cell concentration of 2 × 10^6^ CFU/mL. Aliquots (200 μL) of bacterial suspension were added to the wells of a 96-well plate. An equal volume of nutrient broth medium containing CAN, BBR, B+C and BC solutions (dissolved in DMSO) was added to each well to achieve final concentrations (1/2MIC80 or MIC80). The final concentration of DMSO was 0.1% (*v*/*v*). Nutrient broth medium supplemented with 0.1% DMSO was used as the negative control. The samples were cultured at 37 °C, and the absorbance value was measured at OD620 nm within 24 h or 32 h.

### 2.8. Biofilm Growth

In order to explore the antibacterial mechanism of drugs, the biofilm formation of bacteria was measured. After culturing overnight, the bacterial medium containing *Staphylococcus aureus, Pseudomonas aeruginosa* or *Escherichia coli* were adjusted to an OD620 of 0.1 and then 10 µL of the OD-adjusted bacterial suspension were added to wells of 12-well flat-bottom polystyrene microplates (TCP011012, Biofim, Guangzhou, China), where each well was filled with 1 mL nutrient broth medium contained CAN, BBR, B+C and BC solutions, respectively. All tested drugs were dissolved in DMSO at the final concentrations of 1/2MIC80. The final concentration of DMSO was 0.1% (*v*/*v*). DMSO was served as a control. Biofilms were subsequently allowed to grow at 37 °C for 24 h. Then, the bacterial medium was sucked out and washed twice with PBS. The bacteria in microplates were fixed with paraformaldehyde for 30 min and then dried at 60 °C. The microplates containing bacteria were dyed with 0.1% crystal violet dye solution for 15 min, then sucked out crystal violet solution, washed with PBS and dried again. Ethanol solution (95%, *v*/*v*) were added into the wells and the microplate was shaken to make the adsorbed crystal violet fall off fully. Finally, the absorbance value was measured at OD570 nm.

### 2.9. SDS-Polyacrylamide Gel Electrophoresis (SDS-PAGE)

The effect of tested drugs to Pseudomonas aeruginosa protein level was analyzed by SDS-PAGE. The pretreatment method of *Pseudomonas aeruginosa* is the same as above. CAN, BBR, B+C and BC dissolved in DMSO were added to bacterial suspension to achieve final concentrations of 1/2MIC80, then 200 μL of the bacterial medium was transferred into each well on 12-well plates at 37 °C for 24 h. The samples of *Pseudomonas aeruginosa* were centrifuged at 10,000× *g* for 5 min at 4 °C, then the supernatant was discarded and the bacterial cell pellet was collected and was washed twice with PBS. The lysate was prepared according to the instructions of bacterial protein extraction kit (Add 10 μL protease inhibitor and 10 μL PMSF to 1 mL lysate; The wet weight of bacteria:the volume of the lysate = 1 g:10 mL). The bacteria were ultrasonically lysed and then centrifuged at 10,000× *g* for 20 min, 4 °C. The supernatant was collected and the protein concentration was determined. The protein concentrations of all samples were adjusted to be consistent; the total loading amount is 10 μg, and the loading buffer (25 μL containing 100 mmol/L Tris-HCl pH 6.8, 10% sodium dodecyl sulfate (SDS), 0.5% bromophenol blue, 50% glycerine, 200 mmol/L dithiothreitol (DTT)) was added into 100 μL of samples. The samples were boiled at 100 °C for 10 min and cooled on ice and then analyzed by SDS-PAGE. After electrophoresis, the protein bands were stained with Coomassie Brilliant Blue R-250 and then decolorized to obtain the separated protein bands.

### 2.10. Field-Emission Scanning Electron Microscopy (FESEM)

FESEM was performed as described by Zhou [27], with some modifications. Bacterial cells (~2 × 10^8^ CFU/mL) were treated with CAN, BBR, B+C, BC at concentrations of 1/2MIC80 and incubated at 37 °C for 24 h. The cells were then harvested by centrifugation (4000× *g* 5 min, 4 °C) and rinsed twice with PBS. Bacterial cells were quickly and gently spread onto a small silicon platelet and allowed to air dry in the room temperature (24 °C) for 2 min, followed by in situ fixation of 2% gluteraldehyde coverage in 0.15 M sodium phosphate buffer (pH 7.4) for 10 min, the excess fixative was softly wicked up with the corner of a sheet of filter paper, the platelets were transferred to prefreezing lyophilizer (Alpha 1-2, Martin Christ Company, Germany) at −45 °C and continually dried in situ for 30 h under vacuum. The samples were coated with gold pallatium metal for 80 s at 20 mA and transferred to FESEM (SU8010, HITACHI, Japan) for detections at an accelerating voltage of 5 kV.

### 2.11. Statistical Analysis

GraphPad Prism 8.0 software was used for the statistical analysis. Data were expressed as mean ± standard deviation (S.D.). Differences with statistical significance between groups were calculated by ANOVA followed by Tukey’s post hoc test. *p* < 0.05 was considered statistically significant.

## 3. Results and Discussion

### 3.1. Results of Dihydroberberine

Dihydroberberine: a pale-yellow solid (75.0% yield); ^1^H NMR (see Appendix A, 400 MHz, CDCl_3_) δ 7.17 (s, 1H), 6.73 (s, 2H), 6.57 (s, 1H), 5.93 (t, J = 6.8 Hz, 3H), 4.32 (s, 2H), 3.84 (s, 6H), 3.13 (t, J = 5.6 Hz, 2H), 2.87 (t, J = 5.4 Hz, 2H) [27,28].

### 3.2. Results of CAN Bromide

CAN bromide: a pale-yellow solid (60.0% yield), m.p. 170 °C; ^1^H NMR (see Appendix A, 400 MHz, CDCl_3_) δ 7.42 (s, 2H), 7.29–7.09 (m, 3H), 6.97 (s, 3H), 6.62 (s, 1H), 4.07 (s, 3H), 3.61 (s, 4H), 3.42 (dd, 2H), 2.27 (s, 3H). ^13^C NMR (see Appendix A, 100 MHz, CDCl_3_) δ 163.43, 160.96, 143.05, 141.67, 138.54, 137.01, 135.99, 130.88, 128.87, 127.25, 127.18, 126.15, 125.90, 122.84, 115.93, 115.72, 81.40, 76.54, 53.54, 34.26, 33.67, 30.34, 29.83, 19.39, 15.39. ^19^F NMR (see Appendix A, 300 MHz, CDCl_3_) δ -114.99.

### 3.3. Results of BC

BC: a pale yellow solid (46.0% yield), m.p. 104 °C; ^1^H NMR (see Appendix A, 400 MHz, CD_3_OD) δ 9.83 (s, 1H), 8.88 (s, 1H), 8.15 (d, J = 9.1 Hz, 1H), 7.95 (d, J = 9.0 Hz, 1H), 7.74 (s, 1H), 7.64 (d, J = 16.1 Hz, 2H), 7.54 (dd, J = 7.9, 5.2 Hz, 1H), 7.38–7.29 (m, 1H), 7.21 (t, J = 7.7 Hz, 1H), 7.17–7.07 (m, 2H), 7.04 (s, 1H), 6.12 (s, 2H), 5.70 (s, 1H), 5.02–4.78 (m, 3H), 4.17 (t, J = 6.5 Hz, 3H), 4.04 (d, J = 10.8 Hz, 9H), 3.18–3.12 (m, 3H), 2.23 (t, J = 12.0 Hz, 3H), 2.13 (t, J = 7.4 Hz, 2H), 1.63–1.55 (m, 3H). ^13^C NMR (see Appendix A, 100 MHz, CD_3_OD) δ 177.82, 172.99, 152.12, 151.96, 149.87, 145.71, 139.59, 135.09, 134.94, 134.84, 134.45, 134.34, 131.84, 131.50, 131.37, 131.13, 131.00, 130.94, 128.26, 128.07, 126.11, 124.54, 123.26, 121.80, 121.47, 109.39, 106.53, 103.65, 62.59, 57.66, 57.21, 36.63, 35.37, 35.27, 33.04, 30.91, 30.75, 30.45, 26.73, 23.71, 22.59, 22.55, 14.71, 14.44. ^19^F NMR (see Appendix A, 300 MHz, CD_3_OD) δ −115.39. MS spectrum (FTMS + p ESI Full MS, see Appendix A) *m*/*z* (%): 336.1222 (M-430.1598, 100, calcd. for C_20_H_18_NO_4_ 336.1200); 766.28320 (calcd. for C_44_H_45_O_8_NFS 766.28444). The purity of BC was identified as about 99% by HPLC at 365 nm or 210 nm (see Appendix A).

### 3.4. MIC80 of BBR, B+C and BC against E. coli, S. aureus and P. aeruginosa

BBR, B+C and BC showed antibacterial activity against *Staphylococcus aureus*, *Pseudomonas aeruginosa* and *Escherichia coli* (as shown in Table 1). The minimum inhibitory concentrations of BC against *Staphylococcus aureus*, *Pseudomonas aeruginosa and Escherichia coli*. were 0.2, 0.4 and 0.4 mM, respectively. The MIC80 values of B+C to three strains were 0.5, 0.7 and 0.7 mM, respectively. The MIC80 values of BBR to three bacteria were 0.7, 0.7 and 0.9 mM, respectively. However, we did not obtain the MIC80 of CAN at tested concentrations (<1 mM). Compound BC showed the strongest antibacterial activities and broad-spectrum antibacterial effects, which could inhibit the activities of Gram-positive and Gram-negative bacteria at the same time. The bacteriostatic effect of B+C was better than that of BBR alone, indicating that BBR and CAN have synergistic bacteriostatic effects. Interestingly, antibacterial effects of compound BC were better than that of B+C.

### 3.5. Effects of BC on E. coli, S. aureus, and P. aeruginosa Growth

The suppressive effects of CAN, BBR, B+C, BC on *E. coli*, *S. aureus,* and *P. aeruginosa* were measured at final concentrations of (1/2MIC80 or MIC80). CAN did not have MIC80 and its concentrations involved in following investigations were referred to those of BBR. BC showed more obvious antibacterial effect than other treatment groups (Figure 5a–f). The growth of *P. aeruginosa* with 0.2 mM BC, *E. coli* with 0.2 mM BC or *S. aureus* with 0.4 mM BC did not increase significantly within 24 h or 32 h. The effect of CAN on *P. aeruginosa* over time under the tested conditions had no significant difference compared with the control group (Figure 5a); however, CAN had weak antibacterial activity on the growth of *S. aureus* and *E. coli* at some time points (Figure 5b–f). The other treatment groups including BBR or B+C also showed growth inhibition compared with the control group, but this effect was significantly weaker than that of BC (Figure 5a–f).

### 3.6. Antibiofilm Activity of BC

The antibiofilm potency of BC was also found against the three test strains, which were studied using crystal violet staining (Figure 6a–c). For *P. aeruginosa*, CAN, BBR, B+C and BC showed inhibitory effect on biofilm formation, and the inhibitory effect of BC was the most obvious. For *E. coli* and *S. aureus*, CAN had little effect on biofilm formation, BBR had a weak inhibitory effect on biofilm formation, but B+C and BC groups had obvious inhibitory effects on biofilm formation, and BC group had the best inhibitory effect. Figure 6d shows the results of crystal violet staining the biofilm adsorbed on the pore wall. It could be seen more intuitively that BC has the strongest inhibitory effect. Therefore, it could be speculated that the mechanism of the bacteriostatic effect of BC was to destroy the formation of the bacterial biofilm.

### 3.7. SDS-PAGE Analysis

The effect of tested drugs to Pseudomonas aeruginosa protein level was analyzed by SDS-PAGE as shown in Figure 7. The intracellular soluble protein of *P. aeruginosa* treated with BC was significantly different from the control group. The protein bands after BC treatment were significantly lighter and more blurred, indicating that BC had a destructive effect on the intracellular protein of *Pseudomonas aeruginosa*, so as to achieve the effect of bacteriostasis. The protein bands of BBR and B+C groups were also weakened compared with the control group, but the effect was far lower than that of BC treatment group, indicating that the protein destruction ability of BC for *P. aeruginosa* was significantly better than that of BBR and B+C groups. It seemed that CAN did not inhibit protein synthesis compared with the control group at all. These results suggested that BC could significantly inhibit protein synthesis of *P. aeruginosa*, and the antibacterial mechanism of BC might involve inhibiting protein synthesis and resulting in shallower protein bands.

### 3.8. The Morphology of Bacteria Observed by FESEM

The cell morphological changes observed by FESEM were shown in Figure 8. The surface of untreated bacteria was smooth, without folds and gullies, and the bacteria were complete. Compared with the normal control group, the bacterial cells in BC group showed serious depression, obvious folds on the cell surface and cell damage, indicating that BC caused damage to the bacteria, which was also consistent with the results of SDS-PAGE and biofilm. The bacterial surface treated with CAN showed slight wrinkles, which was not obvious compared with the untreated normal control group, and the antibacterial effect of CAN alone was poor. There were more folds on the surface of the bacteria in BBR group, and the biofilm of the bacteria became uneven and shrunk compared with untreated control group. The bacterial shrinkage of B+C group was deeper than that of BBR group, indicating that the combination of substances exhibits a synergistic antibacterial effect. However, BC seemed to have more obvious folds on the cell surface and more cell damage than B+C, which indicated that anti-bacterial activity of BC was better than that of B+C.

## 4. Discussion

Although BBR has good medicinal value including anti-bacterial, hypoglycemic and so on. However, its wide clinical application is limited due to its relatively low solubility in plasma, poor stability and low bioavailability [29]. It is also reported that BBR has synergistic effect with other drugs and can enhance its antibacterial activity [30]. In addition, the structure of BBR is modified to enhance the antibacterial properties of BBR derivatives. Through the modification of the structure of BBR at different positions, e.g., C8, C9, C12 and C13, its derivatives are obtained to enhance the antibacterial properties of BBR [9]. Increased length of alkyl chain at C13 may increase lipophilicity of BBR derivatives and enhance antibacterial activity. In this study, we reduced dihydroberberine to obtain hydroberberine, and connected berberine at C13 position with CAN bromide (BR-C) to enhance the antibacterial ability of berberine, which was likely mediated by increasing lipophilicity of BBR. Because of the high incidence of infectious diseases caused by drug-resistant bacteria in recent years, these BBR derivatives have very important application value.

As a drug for treating diabetes, CAN not only plays a role in lowering blood sugar, but also has many other effects including cardiovascular protection, anti-inflammatory effects and gut microbes-regulating activities. Diabetes is associated with increased bacterial infection. For these patients, an ideal drug should have both anti-diabetic and anti-bacterial activities. Although CAN has a preferable hypoglycemic effect, its antibacterial activity was not confirmed. Combining with other anti-bacterial compounds or chemical modification might be useful for solving this problem.

Although we can directly combine BBR with CAN instead of chemically binding BBR to CAN, chemical characters, e.g., low bioavailability of BBR, were still not improved fundamentally. A new compound from different units may produce new activities besides existing activities. Therefore, in this study, we wanted to obtain a new compound by connecting BBR and CAN by chemical bond; thus, this compound has stronger activities than BBR or CAN alone or the BBR+CAN combination. Firstly, the hydroxysilyl ether of CAN was brominated, then CAN bromide was directly connected with BBR, and finally the end product BC was obtained. Unexpectedly, we did not find BC showed a comparable hypoglycemic to CAN and indicated BC might change the hypoglycemic chemical structure of CAN (data not shown). Despite this, the results showed that compound BC showed stronger antibacterial ability than BBR, CAN and the combination of BBR and CAN as expected. Compound BC has broad-spectrum antibacterial activity and has good inhibitory effect on Gram-positive bacteria and Gram-negative bacteria. In further research, it was found that compound BC could destroy bacterial morphology and bacterial biofilm, and the compound could also destroy bacterial intracellular proteins and skeletal structure (Figure 9).

In summary, the above results show that compound BC, as a new BBR salt derivative, was successfully synthesized and has better antibacterial activities than BBR alone and even for the BBR and CAN combination. These results indicates that BC has a very wide application prospect in antibacterial treatment, and provides a new scheme for the treatment of drug-resistant bacterial strains. However, exact molecular mechanisms and further clinical application require further investigation.

## Figures and Tables

**Figure 1 molecules-27-02948-f001:**
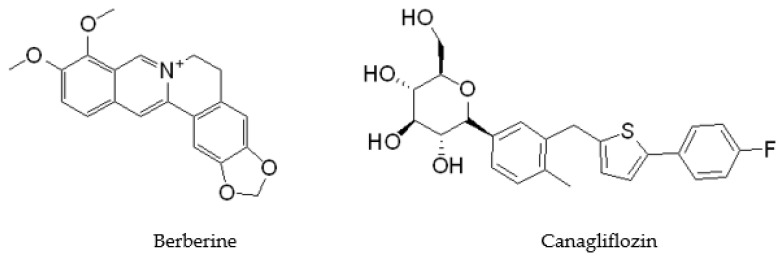
Chemical structure of Berberine and Canagliflozin.

**Figure 2 molecules-27-02948-f002:**
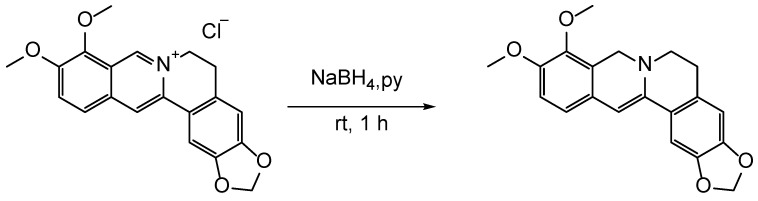
Synthesis of dihydroberberine. py, pyridine; rt, room temperature.

**Figure 3 molecules-27-02948-f003:**
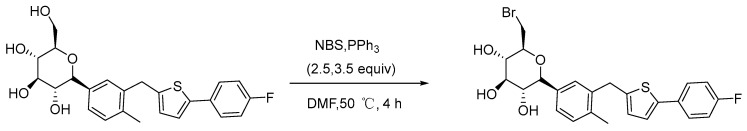
Modification of CAN. Bromination products of hydroxysilyl ether of CAN.

**Figure 4 molecules-27-02948-f004:**
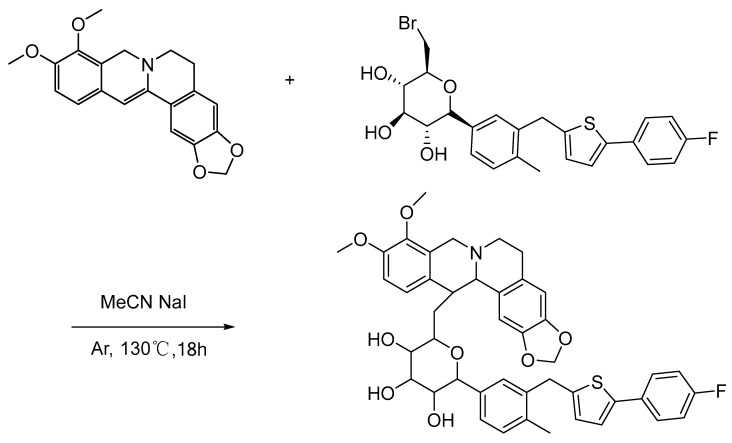
Synthesis of BC. Reagents and conditions: MeCN, NaI, Ar, 130 °C, 18 h.

**Figure 5 molecules-27-02948-f005:**
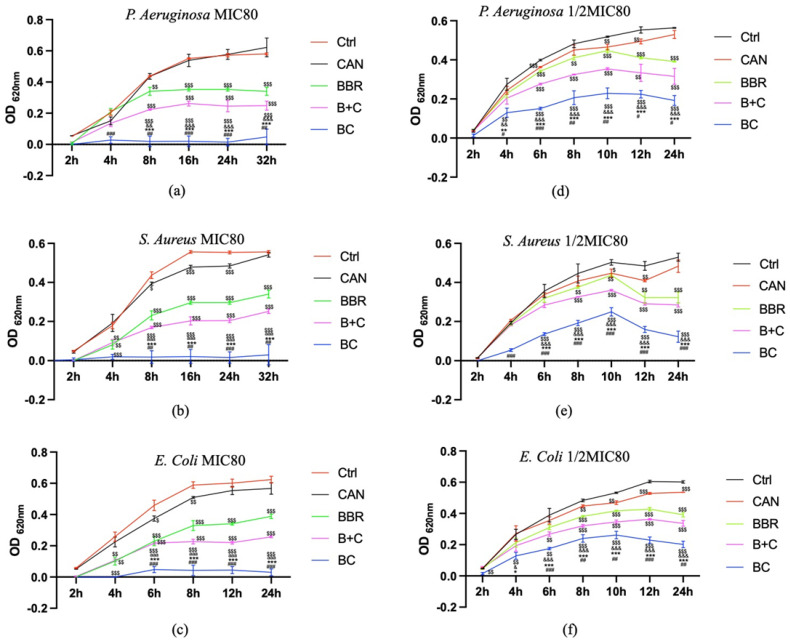
Effects of CAN, BBR, BBR+CAN and BC on *E. coli, S. aureus*, and *P. aeruginosa* growth. (**a**) Antibacterial effect of drugs on *P. aeruginosa* at 0.2 mM; (**b**) Antibacterial effect of drugs on *S. aureus* at 0.4 mM; (**c**) Antibacterial effect of drugs on *E. coli* at 0.4 mM. Changes in the inhibitory effects of (**d**) *P. aeruginosa*; (**e**) *S. aureus*; (**f**) *E. coli* with time in response to CAN, BBR, B+C and BC at 1/2MIC80. ^$^ *p* < 0.05, ^$$^ *p* < 0.01, ^$$$^ *p* < 0.001 vs. Ctrl; ^&^ *p* < 0.05, ^&&^ *p* < 0.01, ^&&&^ *p* < 0.001 vs. CAN; * *p* < 0.05, ** *p* < 0.01, *** *p* < 0.001 vs. BBR. ^#^ *p* < 0.05, ^##^ *p* < 0.01, ^###^
*p* < 0.001 vs. B+C. Data were shown as mean ± S.D. (n = 3).

**Figure 6 molecules-27-02948-f006:**
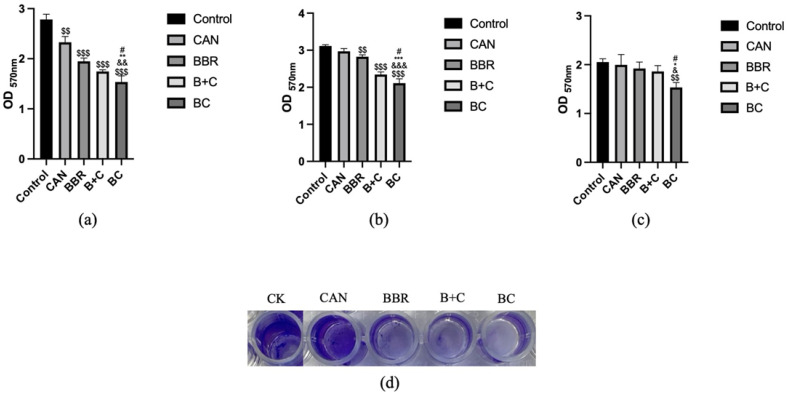
Inhibitory effects of CAN, BBR, and BBR+CAN. BC on bacterial biofilm formation with the concentration of 0.1 mM for 24 h. The bacterial strain: (**a**) *Pseudomonas aeruginosa*; (**b**) *Staphylococcus aureus*; (**c**) *Escherichia coli*; (**d**) Biofilm formation of *Pseudomonas aeruginosa.* Data are expressed as mean ± SD (n = 3); ^$$^ *p* < 0.01, ^$$$^ *p* < 0.001 vs. Ctrl; ^&^ *p* < 0.05, ^&&^ *p* < 0.01, ^&&&^ *p* < 0.001 vs. CAN; * *p* < 0.05, ** *p* < 0.01, *** *p* < 0.001 vs. BBR. ^#^ *p* < 0.05 vs. B+C.

**Figure 7 molecules-27-02948-f007:**
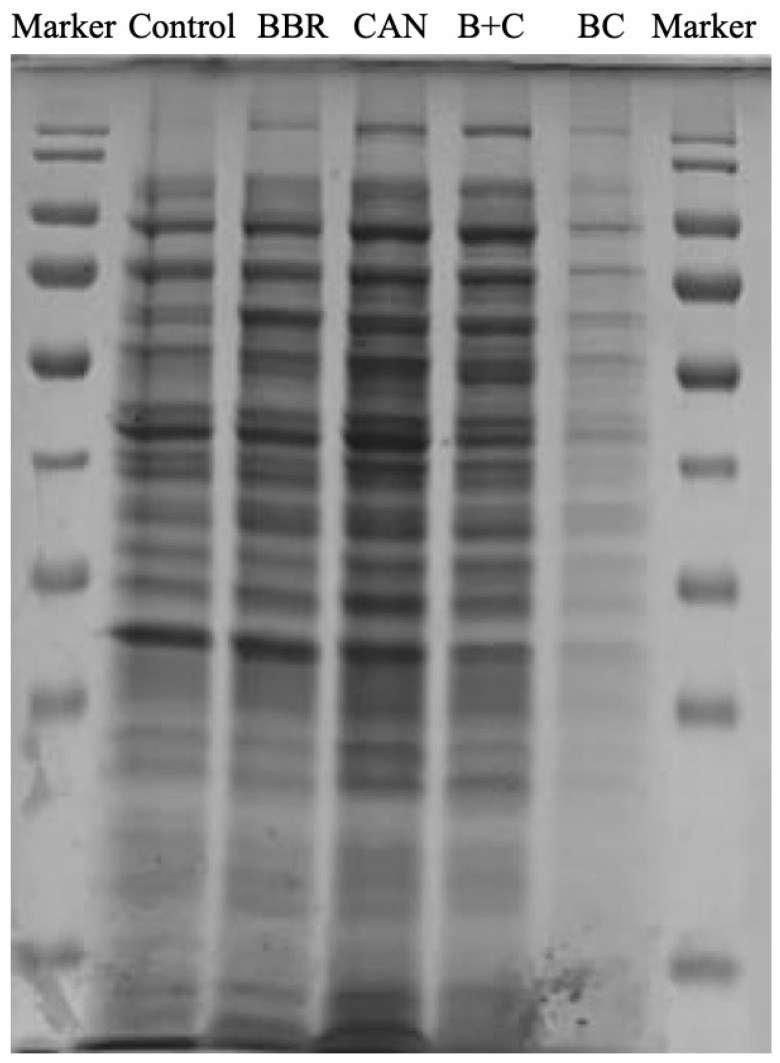
SDS-PAGE analysis of intracellular soluble proteins of *Pseudomonas aeruginosa* treated with BC of 0.1 mM for 24 h.

**Figure 8 molecules-27-02948-f008:**
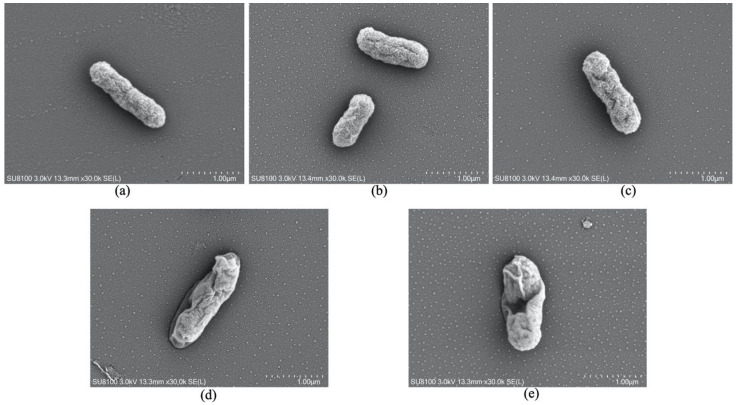
Field-emission scanning electron micrographs of *Pseudomonas aeruginosa*. (**a**) Untreated control cells at 24 h post-inoculation; (**b**) Bacterial cells treated with CAN at 1/2MIC80 (0.1 mM) for 24 h; (**c**) Bacterial cells treated with BBR at 1/2MIC80 (0.1 mM) for 24 h; (**d**) Bacterial cells treated with B+C at 1/2MIC80 (0.1 mM) for 24 h; (**e**) Bacterial cells treated with BC at 1/2MIC80 (0.1 mM) for 24 h.

**Figure 9 molecules-27-02948-f009:**
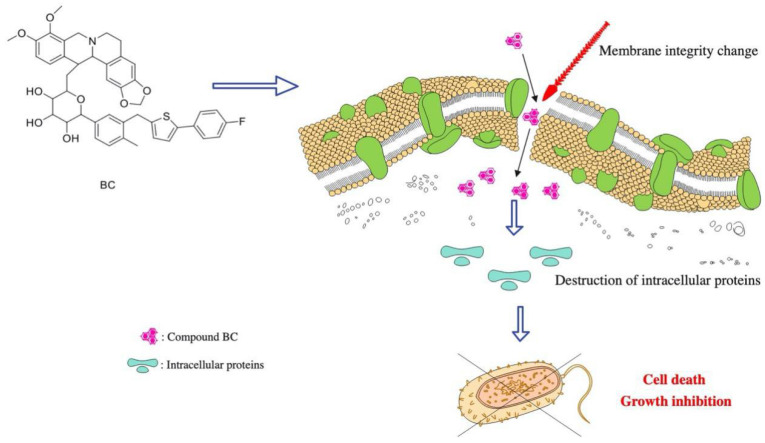
Bactericidal mechanism of compound BC.

**Table 1 molecules-27-02948-t001:** Minimum inhibitory concentrations (MIC80) of BBR, BBR+CAN and BC against different strains.

Bacterial Strain	Source	Drug	MIC80 (mM)
*Pseudomonas aeruginosa*	BNCC337005	BBR	0.71 ± 0.02
		BBR+CAN	0.53 ± 0.06
		BC	0.22 ± 0.04 **^,#^
*Staphylococcus aureus*	BNCC186335	BBR	0.93 ± 0.04
		BBR+CAN	0.74 ± 0.02 *
		BC	0.38 ± 0.06 **^,#^
*Escherichia coli*	BNCC337004	BBR	0.91 ± 0.02
		BBR+CAN	0.73 ± 0.02 *
		BC	0.39 ± 0.04 **^,#^

Data are shown as mean ± S.D. (n = 3). * *p* < 0.05, ** *p* < 0.01 vs. BBR; ^#^
*p* < 0.05 vs. BBR+CAN.

## Data Availability

All data are available in a repository or online in accordance with funder data retention policies.

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
