# Peer review of "Synthesis of Berberine and Canagliflozin Chimera and Investigation into New Antibacterial Activity and Mechanisms"

_molecules, 2022, doi:10.3390/molecules27092948_

Round 1

Reviewer 1 Report

The remarks on the manuscript are listed below from least to most significant.

  1. line 53: The empirical formula of the cation is written, so you need to indicate that this is a charged particle
  2. Berberine and canagliflozin are repeatedly mentioned in the introduction, the appearance of these structures in the text before they appear in the article is sorely missed
  3. It is more correct to call hydroberberine dihydroberberine, because the degree of reduction of ring C can be different
  4. line 190: alcohol, apparently, is indicated by mistake. The reaction is carried out in DMF
  5. Sections 3.1 -3.3 do not indicate the amount of solvents per a certain amount of substance, it must be indicated. Here you also need to specify the ratio of solvents in the eluent system
  6. In paragraphs 3.2 and 3.3, MS and EA data are missing
  7. line 194: dilute the filtrate with water and extract with diethyl ether?
  8. lines 111, 120 and 132: indicate the amount of DMSO, be sure to indicate the content of DMSO in the final solution
  9. line 208: is the solution in acetonitrile applied to silica gel? Or pre-evaporate acetonitrile?

10 MICs throughout text better corrected to MIC80

  1. line 312: bad senence: “the synergistic antibacterial effect of the two drugs was greater than that of BBR alone”. Or: the antibacterial effect of a combination of substances is greater, better than the sum of the antibacterial effects of each. Or: the combination of substances exhibits a synergistic antibacterial effect
  2. line 314: However, BC seemed to have damages than the other tested drugs including B+C. - the sentence doesn't make sense
  3. lines 324-328. "In addition, the structure of BBR is modified to enhance the antibacterial properties of BBR derivatives. In addition, there are some reports on the structural transformation of BBR. Through the transformation of the structure of different positions of BBR, its derivatives are obtained to enhance the antibacterial properties of BBR [16]." - a generally meaningless set of sentences that should be rewritten into a brief overview of the modifications of berberine, leading to increased antibacterial activity. With an emphasis on 9-O-derivatives.
  4. Literary references are absolutely not formalized and often do not correspond to reality. For example, the above reference 16.
  5. A number of conceptual remarks: it is not clear from the introduction why a structural fragment, pharmacophore for diabetes, was introduced to enhance antibacterial activity. And only at the end of the text of the entire article it is mentioned that the expected hypoglycemic effect was not found. An article will look more advantageous if it is written more honestly. Namely: to point out that the introduction of pharmacophores is a fashionable and frequently used trend, they used this approach, but instead of increasing the hypoglycemic properties, they got an increase in antibacterial properties (which, by the way, has already been observed for 9-O-derivatives of berberine)

Reviewer 2 Report

The manuscript entitled “Synthesis of Berberine and Canagliflozin chimera and investi-2 gation into new antibacterial activity and mechanisms” reported excellent study, however before accepting the manuscript few corrections are required as mentioned below:

  1. Elaborate the rationality behind the connected the two compounds BBR and CAN? Millions of compounds have already reported which can also be used. Authors should discuss the logic behind the use of only these two compounds.
  2. Page 2: Line 81: Bacterial names should be italic throughout the manuscript.
  3. Page 2: Line 88: Compound synthesis portion is not clear: I think authors should discuss the synthesis scheme of BC.
  4. Page 4: Line 147: check centrifuge reading unit.
  5. Page 4: Line 198: Correct temperature symbol (º).
  6. Page 4: Line 180, 188, 205: Authors reported synthesis details in Results and Discussion, it should be include in Materials and methods.
  7. For comparison antibacterial activity, authors can choose marketed antibiotics.
  8. The ±SD can also be reported for MIC in Table 1.
  9. In Figure 3: All bacterial names should be italic.
  10. Page 10: Line319: Do correct heading Discussion.
  11. Check all reference in referencing section.

Reviewer 3 Report

The synthetic protocols including schemes and details of experiments need improvement; The purity of final product (BC) is not clear from NMRs and HRMS, probably is low even though there is product signal from HRMS but the scanned scope is very narrow. The style of references is messy. So not sure the reliability of bioassays.

Reviewer 4 Report

The manuscript “Synthesis of Berberine and Canagliflozin chimera and investigation into new antibacterial activity and mechanisms” by Hao et al. introduces a synthesis of novel berberine derivate bearing Canagliflozin moiety and presents interesting findings on its antibacterial activity against several bacterial strains. Overall, the manuscript is well written and can be accepted after minor changes.

Suggestions to the Authors:

Line 23: Should read: However, due to low lipophilicity of berberine, it is …

Lines 25-26: Please reformulate this sentence to better reflect that the ultimate goal of the study was to prepare new chimera of berberine and Canagliflozin with higher antimicrobial activity that single agents.

Lines 34-35: Why is the term “kaglitazin” used here? Wouldn’t it be better like this: “The minimal inhibitory concentration of compound BC on three bacteria was significantly better than that of berberine and Canagliflozin alone and the combination of berberine and Canagliflozin.

Line 56: “fat solubility” replace with “lipophilicity”.

Line 81: Indicate here which bacteria is Gram-positive and which is Gram-negative.

Lines 142-143: Should read: The effect of tested drugs to Pseudomonas aeruginosa protein level was analyzed by SDS-PAGE.

Line 204: Should simply read: Structure of Canagliflozin bromide.

Lines 264-265: Check and correct x-axis of Figure 3c.

Lines 271-285: Statistical analysis should be corrected the way to show the statistical significance of antibacterial effects of single agents (B, C), their combination (B+C), and BC compare to control (untreated) bacteria.

Lines 297-299: Specify the meaning of NOR (2nd lane of Figure 5).

Figure 5: The authors might consider to check and show the effect of 0.1 mM B, 0.1 mM B+C and 0.1 mM BC on protein level of Pseudomonas aeruginosa in time, meaning how the protein levels look like after treatment of bacteria for 2h, 4h, 8h, 12h and 24h. This might allow to find out how long it takes the new BC derivate to affect the protein level of bacteria/damage cell membrane and thus demonstrate its improved antimicrobial potential.  

Round 2

Reviewer 1 Report

Just 2 small remarks:
1) please check the subsection numbering in section 2
2) line 154: it is quite enough to represent the ratio of solvents as 1:1:0.02 instead of "Vmethylene chloride:Vmethanol:Vtriethylamine=1:1:2%"

I apologize to the authors for the remark about the need to emphasize the 9-O-derivatives. This is my omission, I inattentively looked that the article is devoted to the synthesis of the derivative at position 13.

Author Response

Thank you for your reminder, we have done that in the newly revised manuscript.